# Cyclic Test Time Augmentation with Entropy Weight Method

**Sewhan Chun**[1]  **Jae Young Lee**[2]  **Junmo Kim**[2]

[1]NAVER CLOVA, Republic of Korea
[2]SIIT Lab, KAIST, Republic of Korea

## Abstract

In the recent studies of data augmentation of neural networks, the application of test time augmentation has been studied to extract optimal transformation policies to enhance performance with minimum cost. The policy search method with the best level of input data dependency involves training a loss predictor network to estimate suitable transformations for each of the given input image in independent manner, resulting in instance-level transformation extraction. In this work, we propose a method to utilize and modify the loss prediction pipeline to further improve the performance with the cyclic search for suitable transformations and the use of the entropy weight method. The cyclic usage of the loss predictor allows refining each input image with multiple transformations with a more flexible transformation magnitude. For cases where multiple augmentations are generated, we implement the entropy weight method to reflect the data uncertainty of each augmentation to force the final result to focus on augmentations with low uncertainty. The experimental results show convincing qualitative outcomes and robust performance for the corrupted conditions of data.

## 1 INTRODUCTION

Study of test time augmentation (TTA) is a field of data augmentation, which involves transforming an input image to augment different forms of itself for neural network prediction during the test time. This generates multiple softmax outputs, which can be integrated by averaging them to extract the final single output. Such a method has been known to result in more robust and better performance from the neural networks [Krizhevsky et al., 2012, Ashukha et al., 2020]. Conventionally, which transformations to use are heuristically set in global-level (*i.e.* performing the same types of augmentation to all the input data) for the domain.

However, there are limitations for conventional TTA. The major concern is the cost. TTA policy refers to the scheme of how many augmentations of what transformations with what magnitude for each augmentation would be utilized [Molchanov et al., 2020]. While increasing the number of augmentation in the policy usually results in better performance, the cost requirement has to increase in a multiplicative manner. Because of such poor cost efficiency, many TTA applications can be found in tasks where accuracy plays an important role, such as artificial intelligence competitions and medical or biological image processing [Krizhevsky et al., 2012, Perez et al., 2018, Matsunaga et al., 2017, Moshkov et al., 2020].

Another concern involves the inflexibility of the policy. While the suitable policy should maintain intra-class invariance (*i.e.* invariance of the label under transformation) and inter-class distinctiveness (*i.e.* ability to maintain distinctive features to distinguish between classes) of input data to the model [Sato et al., 2015, Shanmugam et al., 2020], in conventional scheme, the policy is found heuristically and applied in global-level. This could bring disruption and inconvenience to the policy establishment. For example, a horizontal flip is known to be a common and effective TTA transformation with intra-class invariance and inter-class distinctiveness for most images from ImageNet dataset [Krizhevsky et al., 2012, Deng et al., 2009]. However, from MNIST dataset [Deng, 2012], while visually symmetric numbers ("1","8") could be acceptable to such transformation, orientation-sensitive images ("7","6","2","5") could lose their intra-class invariance and inter-class distinctiveness from the flipping, losing features to classify them as their original labels.

To overcome such limitations, trainable TTA policy search methods were introduced. These approaches aim to structure the most suitable TTA policy as an optimization problem, finding the most helpful augmentations from various candi-

*Accepted for the 38th Conference on Uncertainty in Artificial Intelligence* (UAI 2022).

dates of transformations and their magnitudes. From Greedy Policy Search (GPS) [Molchanov et al., 2020], multiple augmentations can be generated in the policy, where each augmentation is regarded as a sub-policy, capable of consisting of multiple transformations with corresponding magnitudes. While GPS has a global-level TTA scheme, some of the studies aim to find more specific levels of data dependency of TTA policy, namely class-level and instance-level (*i.e.* applying transformations to the input image depending on individual input data condition).

Trainable TTA policy has also contributed to the robustness of neural network prediction. Contrary to the promising performance of neural networks, it has been studied that they could be vulnerable to perturbations or corruptions in data [Goodfellow et al., 2015, Hendrycks and Dietterich., 2019]. Many studies in data augmentation methods have achieved strong robustness [Dan et al., 2020, Cubuk et al., 2019, Lim et al., 2019] against the damages. Previous works [Kim et al., 2020, Molchanov et al., 2020] showed that TTA could also improve the robustness. With a suitable TTA policy, corruption in the image could be suppressed by modifying the test image directly via suitable transformations. Kim et al. [Kim et al., 2020] has recently introduced the first instance-level TTA policy search method, where which transformation to proceed is determined by the condition of each instance of input image. With the application of a loss predictor, their work was able to achieve robustness improvement with only a small amount of additional computation cost.

In this work, we introduce cyclic TTA with entropy weight method (EWM) in classification task by implementing multiple transformations and reflecting uncertainty directly to each prediction result from augmentations. As we follow that the instance-level TTA is the effective level of the data dependency, we believe that there is more potential room for improvement to the loss prediction pipeline [Kim et al., 2020] in terms of flexibility. With an iterative maneuver of the loss predictor, each image can be assigned with multiple transformations with a more flexible magnitude. For multiple augmentations case, we also introduce the implementation of modified EWM to attenuate the softmax output with high data uncertainty. Because the cost for the calculation of the entropy is relatively minor, the EWM can easily be adapted to improve the robustness of network prediction.

## 2 RELATED WORKS

**Test time augmentation:** TTA for neural network prediction has been used for a while. Many innovative neural network performances on ImageNet dataset [Deng et al., 2009] used TTA method [Krizhevsky et al., 2012, Szegedy et al., 2015, Simonyan and Zisserman, 2015, He et al., 2016] for their records, using augmentations of numerous cropped patches from the original images. This helped to result in accuracy improvement with the multiplicative cost increase.

For TTA's capability to directly modify the data during test time, TTA has been studied to possess more potentials, such as uncertainty estimation to data distillation [Wang et al., 2019, M.S. and Berens, 2018, Radosavovic et al., 2018]. TTA policy search is one of the attempts to find the solution to the cost limitation and further improvement of its effectiveness. Sato et al. [Sato et al., 2015] were one of the first to analyze TTA policy, building an optimal decision rule to achieve improvement in generalization. GPS [Molchanov et al., 2020] was introduced as a learnable global-level TTA policy search method, greedily building a global-level policy. GPS showed excellent improvement in accuracy and robustness, performing multiple transformations with flexible magnitude to each sub-policy. Shanmugam et al. [Shanmugam et al., 2020] proposed a TTA policy with class-level data dependency. Their work involves training a set of parameters to learn the relation between class and each augmentation and to use it as a post-processing method to extract one final prediction from the multiple predictions. Recently, Kim et al. [Kim et al., 2020] proposed a TTA policy with instance-level data dependency. They had trained a loss predictor to be capable of predicting which transformation would be suitable for the target network (*i.e.* the main classifier used for the task). Their contribution is that such a pipeline is very cost-efficient, even with a single suitable augmentation could increase the robustness effectively. However, unlike GPS, such a pipeline could not implement multiple transformations with flexible magnitudes on a single image, each sub-policy to intake a single transformation from a set of predefined transformations.

**Robustness to Corruption:** While modern neural networks achieve high performance, exceeding human capabilities, many studies show that they can easily malfunction for corruptions and perturbations from various sources from real-life implementations [Goodfellow et al., 2015, Hendrycks and Dietterich., 2019]. Hendrycks et al. [Hendrycks and Dietterich., 2019] introduced a benchmark for corruptions with ImageNet data, namely ImageNet-C, simulating 19 different types of corruption for network robustness evaluation. Many data augmentation approaches [Dan et al., 2020, Cubuk et al., 2019, Lim et al., 2019] were introduced to enhance the robustness, resulting in significant improvement for various kinds of corruptions.

**Uncertainty Estimation:** Uncertainty estimation acts as an indicator for the confidence of network prediction. Many applications in deep learning involve implementation of the uncertainty to provide additional information for the final prediction [Gal, 2016]. In the field of active learning, where uncertain data are queued to be labeled from a set of unlabeled data, a loss value can be used as a means for estimation of the uncertainty. In the case of active learning classification [Yoo and Kweon, 2019], a separate loss predictor module can be trained to estimate expected loss magnitude with much less cost, providing a faster and more

efficient method to find samples with high expected loss value to queue and select the uncertain unlabeled data. In this case, the loss value can be regarded as an indication of how uncertain the data is for the target network (the classifier), for samples with high loss values bring relatively major change to the condition of the neural network.

From another point of view, according to the previous study [Malinin et al., 2020], overall uncertainty measurement from neural network prediction can be divided into knowledge uncertainty and data uncertainty. In this paper, we focus on the data uncertainty, irreducible uncertainty due to the nature of complexity or noise in the data. In the classification task, data uncertainty can be calculated as the expected entropy value of softmax outputs. The expectancy can be calculated by averaging the entropy values from multiple predictions (softmax outputs) by multiple models from an input data.

**Entropy Weight Method:** In the field of decision making, EWM is used to reflect the degree of disorder of a system [Amiri et al., 2014, Liu et al., 2010]. Many studies in water quality assessment use EWM to reflect the uncertainty among the samples to diminish the importance weights of uncertain assessment parameters. The weights indicate the importance of parameters for the quality assessment and are calculated to be large for low entropy and vice versa. For example, a type of substance (*i.e.* a parameter) detected with a uniform amount from the majority of samples would gain less weight than other parameters, due to the high entropy from the uniformity. With some adaptive modification from previous EWM, we observe that a network prediction might be similarly regarded as the sample from the field of decision making. By reflection of the entropy to the network predictions, we seek to improve the robustness of the predictions with only a small extra cost to calculate the entropy.

# 3 METHOD

Our method includes cyclic modification of the loss prediction pipeline and implementation of the EWM. In section 3.1, we introduce our baseline, the previous loss prediction pipeline illustrated in Figure 1, and the modifications for our method. The cyclic application of the loss predictor will be explained in section 3.2. The iterative manner of transformations tries to find an optimal condition for a given input image. Compare to the previous work, such application contributes to additional flexibility of transformations in TTA policy. The difference between the former method is illustrated in Figure 2. In section 3.3, the modifications and implementation of EWM are explained. In case of multiple augmentations case, where more than one augmentation are used for TTA policy, we aim to reflect the data uncertainty to each augmentation. For uncertainty estimation, we refer to the well-stated definition of the uncertainty by Malinin et al. [Malinin et al., 2020], considering the entropy from

softmax output could represent the data uncertainty (with a difference in that we only use a single network prediction to calculate the data uncertainty).

## 3.1 LOSS PREDICTION FOR THE TRANSFORMATION ESTIMATION

Kim et al. [Kim et al., 2020] introduced an innovative loss prediction pipeline for instance-level image augmentation during test time. As presented in Figure 1, a loss predictor aims to find a suitable transformation among predefined transformations for an input image to be prepared for the target network (*i.e.* classifier). During the test time, an input image is resized and evaluated by the loss predictor. The loss predictor predicts the expected losses for each of the presumable target network predictions with transformed augmentations from the predefined set of transformations. In other words, the loss predictor tells of what transformation would result in the best outcome for the target network, as the least predicted loss value would refer to the transformation with the best condition. The transformation corresponding to the minimum predicted loss is selected as the top 1 choice for the sub-policy. In the case of a single augmentation, such a pipeline guides an input image to go through the suitable transformation, making the classifier to predict from the transformed condition of the image.

**Training the loss predictor:** Training the loss predictor requires the target network to make predictions with an input image in multiple augmented forms in the manner of predefined transformations. During the training, the target network is frozen, only making predictions. For each prediction, cross-entropy loss values from the multiple augmented images are calculated. The loss values from the augmentations are softmax normalized and are fed to the loss predictor as the target values, as Spearman correlation ranking loss [Engilberge et al., 2019]. Ultimately, the loss predictor learns to find which transformation is required to result in the smallest loss value, as the image is evaluated by the target network. Being able to predict with suitable transformation to extract the smallest loss value, the input image has more chances to be classified with the correct answer.

For training the loss predictor on the ImageNet dataset, training data used for training the target network are reused. Although training the loss predictor with a separate validation set seems to be more suitable, for the loss values by the target network prediction from training data would not perfectly simulate the actual test condition, regardless, it has been reported that they do not make much difference in performance.

Additionally, in order to build robustness to corruptions, random sequences of corruption, from the previous study by [Hendrycks and Dietterich., 2019], were given to the input

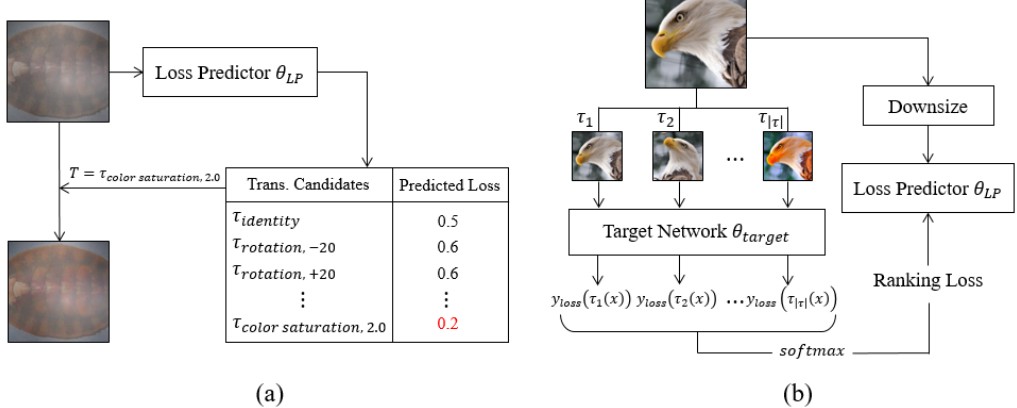

(a)                           (b)

Figure 1: Illustration of the loss prediction pipeline [Kim et al., 2020]. (a) Loss prediction for which transformation $T$ to take on the corrupted image of "chiton" during testing. $\tau_{a,b}$ indicates the predefined transformation of type $a$ with its magnitude $b$. (b) Training algorithm of the loss predictor $\theta_{LP}$. During the training, an input image $x$ is transformed into all of the predefined transformations $\tau$ to produce loss values $y_{loss}(\tau(x))$ by making predictions with the target network $\theta_{target}$. These loss values are given to the loss predictor $\theta_{LP}$ as target values after softmax normalization and as Spearman correlation ranking loss [Engilberge et al., 2019]. The loss predictor intakes the resized input image to learn the correlation between the target network results from the transformed images and the downsized original image condition.

images, simulating various types of real-life conditions of the images.

**Loss predictor architecture:** For the network architecture of the loss predictor, EfficientNet-B0 [Tan and Le., 2019] is used as the backbone. Architectural modifications were taken to utilize multi-level features of input as the active learning loss predictor [Yoo and Kweon, 2019]. The loss prediction pipeline is stated to be cost efficient because the cost for the loss prediction with such a network architecture is relatively negligible to that of the target networks used for the classification [Kim et al., 2020]. Downsizing the image into 64 by 64 pixels has allowed such cost efficiency and aimed for the loss predictor to learn low level features as well.

**Transformation candidates:** As for the predefined transformations, in our method, we have modified the transformation magnitudes to simulate more flexible outcome. The types of transformation include: Identity, Rotation, Zoom, Auto Contrast, Blurring, Sharpening, and Color Saturation. Including the magnitude configurations for each transformation, our method composes 12 different transformation candidates. In Appendix A, we explain the details about the transformations. Overall, the loss predictor suggests one of these transformations with the least expected loss value, which then the transformations takes place to be ready for the target network prediction.

**Multiple augmentations:** In the case of $k > 1$ number of augmentations are used, the top $k$ transformations from the loss predictor suggestion are selected to generate the corresponding $k$ number of augmentations. In case of not

using the EWM, classification results from the augmented images are integrated in a conventional manner, averaging the softmax outputs.

## 3.2 CYCLIC TTA

**Cyclic loss prediction:** Contrary to the former study, our work utilizes the loss predictor in a cyclic manner as shown in Figure 2. Once the image is transformed according to the prediction by the loss predictor, instead of being directly processed by the target network, the modified image is again fed to the loss predictor, forming a cycle. The image goes through the cycle continuously, until the exit signal is activated. We set two conditions for the exit signal to be activated. The first is when the loss predictor predicts the input image should perform identity transformation. This indicates that the image no longer requires additional transformations to result in better condition, ideally presuming an optimal condition of the image. The second condition is when the number of cyclic iteration reaches the predefined hyper parameter of maximum number of the iteration. Because our loss predictor is not perfect to predict the suitable transformation, to prevent rarely happening unbounded continuity of the cyclic loss predictions, we set certain limitation to the number of cycle the loss predictor iterates. Such simple modification can expand the transformation space into a much larger volume of possible combinations from the set of predefined transformations. Given that $T$ and $m$ refers to the number of transformation candidates and the maximum number of iteration respectively, transformation space in our method can be written as $T^m - T^{m-1} + 1$.

While our baseline had $m = 1$ to have the $T$ number of transformation possibilities, it can be shown that larger $m$ in our method opens for more potential candidates for the input image to be transformed into.

For a severely corrupted input image, a single iteration of transformation might not be sufficient to suppress the corruption. For example, if an image should be corrupted by a severe Gaussian noise, following the former method, a blurring transformation would be selected and performed to remove the noise. However, it is possible to leave the residual noise component, for the magnitude of the transformation is predefined and only performed once. On the other hand, cyclic iterations of transformation could continuously try to remove the noise until the loss predictor predicts the condition of the image to be well suited for the classification. In such behavior, it is possible for the cyclic TTA to provide more flexible and multiple types of transformation maneuver as a preprocessing for the task.

Training the loss predictor for cyclic TTA involves dealing with multiple number of corruptions to the input data. The input data are applied with multiple number of corruptions, with similar behavior as the loss predictor from [Kim et al., 2020], the loss predictor is trained to predict what transformation could suppress the corruptions and to result in the least expected loss.

**Multiple augmentations:** In case of $k > 1$ augmentations are to be used, we prepare $k$ number of original images to be processed. In the first iteration of $t = 1$, each image is transformed according the top $k$ transformations from the loss prediction respectively. Starting from the second iteration, unless the exit signal is activated for each augmentation, each image will proceed as normal cyclic behavior, each selecting the top 1 suggestion from their each loss predictions. In short, each of $k$ augmentations starts with different transformation at $t = 1$ and proceeds the cyclic TTA independently. Ideally, if the loss predictions were to be very accurate, all $k$ transformed images would present similar features, assuming that there is only one optimal condition of the input image to be prepared for the classification. In the end, $k$ number of target network predictions are generated as softmax outputs. Assuming the EWM is not used, these are averaged to extract a final prediction for each input image.

**Cyclic TTA cost:** As previously mentioned, the cost for the loss prediction is relatively trivial to that of the target network prediction. For example, our experiments on ImageNet involves a target network takes 4.1 GFLOPs, whereas loss predictor with downsized input image only requires 2.6 MFLOPS. Although our cyclic loss prediction requires multiple iteration of the loss prediction and transformation, because the number of iteration can be controlled with a hyper parameter of maximum number of iteration and the cost of the loss prediction is relatively small, such pipeline

can sustain somewhat similar cost efficiency compare to that of our baseline.

## 3.3 ENTROPY WEIGHTED SUMMATION

**Average integration:** In conventional case of using multiple augmentations for TTA, the integration of the softmax outputs is performed by averaging them. In case of classification task with $n$ classes and $m$ augmentations are used, conventional method to extract the final prediction score for class $j$ ($\leq n$) can be calculated as

$$p_{final_j} = \frac{1}{m} \cdot \sum_{i=1}^{m} p_{i,j} \ , \qquad (1)$$

where $i$ represents the augmentation index and $p_{i,j}$ indicates softmax output element of class $j$ from augmentation $i$. Then, the final classification is decided by choosing the class index $j$ with the maximum value of $p_{final_j}$. Such integration implies weighting each prediction with the same importance. On the contrary, we see that certain augmentations can be more important to provide correct prediction [Shanmugam et al., 2020]. For example, for cropping multiple image patches from an original input image, augmentations can be generated each with a different view. Certain patches might not contain essential features, for parts of the original image could be excluded from cropping. In this case, considering these augmentations as the same importance as the others with more correct information could bring disturbance to the final prediction. The illustrations of such cases are present in Appendix C.

**EWM integration:** Inspired by previous works from the field of decision making [Amiri et al., 2014, Liu et al., 2010, Zhu et al., 2020], we state that each prediction made by corresponding augmentation can be regarded as a sample data with a probability distribution for which decision to make with corresponding implicit uncertainty. We modify the previously established EWM to implement in the neural network prediction. While EWM calculates the entropy among samples of data to calculate weights for evaluation parameter, we calculate the entropy $E_i$ of augmentation $i$ as

$$E_i = -\sum_{j=1}^{n} p_{i,j} \cdot \ln p_{i,j} \ . \qquad (2)$$

The entropy is then used to extract the weight $w_i$ with softmax normalization for each augmentation $i$:

$$w_i = \left( \frac{e^{E_i}}{\sum_{i=1}^{m} e^{E_i}} \right)^{-1} . \qquad (3)$$

By having the reciprocal of softmax entropy to calculate weights, each weight represents how much each augmentation is certain for its prediction. As for the integration of the

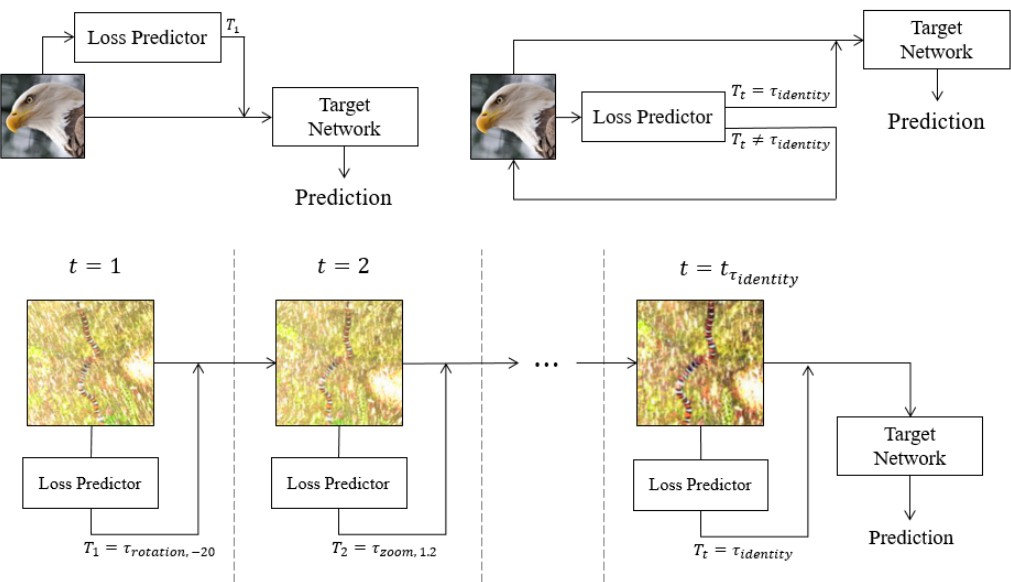

Figure 2: **Top:** Comparison between the previous method (**left**) and the cyclic (**right**) loss prediction pipeline. $T_t$ indicates the suggested transformation at iteration $t$. **Bottom:** Expanded illustration of the cyclic loss prediction. The input image of a "king snake" is corrupted with snow corruption. The image goes through iterative loss prediction cycles until it meets the exit signal. $t_{\tau_{identity}}$ indicates the iteration when the loss predictor suggests identity transformation, which is an exit signal.

predictions from the $m$ augmentations, the final prediction score for class $j$ element is calculated as

$$p_{final_j} = \sum_{i=1}^{m} w_i \cdot p_{i,j} \ . \tag{4}$$

As same as the conventional method, final classification result is done by choosing the class index with the maximum value of $p_{final_j}$. Considering the definition of the data uncertainty by Malinin et al. [Malinin et al., 2020], modified EWM can be regarded as the reflection of data uncertainty to each augmented data, focusing more on less uncertain augmentation and vice versa. Ideally, calculation for more accurate level of data uncertainty involves using more than one neural network. Regardless, with such reflection of the uncertainty, our experiments show that the network prediction can extract more robust predictions to the corrupted data in case of using multiple number of augmentations.

## 4 EXPERIMENTS

### 4.1 IMAGENET CLASSIFICATION

We experiment the effect of cyclic behavior of the loss predictor and EWM on ILSVRC 2012 dataset [Deng et al., 2009]. ImageNet contains 1.2 million images with 1000 classes of real life objects. In addition to the clean condition of the data, we also evaluate our method on ImageNet-C dataset [Hendrycks and Dietterich., 2019], where various types of corruption are simulated with 5 different severity.

The corruptions from the ImageNet-C include 19 different types of algorithmically generated corruptions from noise, blur, weather, digital, and extra categories. While typical error rate is used for the evaluation in clean data, to evaluate the robustness of neural network performance, mean corruption error ($mCE$) metric is used [Hendrycks and Dietterich., 2019]. Overall, in order to evaluate a single iteration of $mCE$, 50,000 (ImageNet validation data size) $\times$ 5 $\times$ 19 samples with size of $224 \times 224$ are used.

In Table 1, we show performance with using ResNet-50 [He et al., 2016] as the target networks for the pipeline. The networks are trained in two different fashions: standard and Augmix [Dan et al., 2020]. Performances from each data augmentation are presented. For comparison, the typically used TTA methods are selected (the typical TTA methods are described in detail in Appendix A). These methods are widely and frequently used conventional TTA methods shown to improve accuracy in many cases [Krizhevsky et al., 2012, Szegedy et al., 2015, Simonyan and Zisserman, 2015, He et al., 2016]. Additionally, we compare our methods to the previous method [Kim et al., 2020], making a single transformation prediction for each image. For each test case, relative costs are presented. These costs only concern the computation load for the classification, for the load for transformations and loss prediction is relatively menial. For the integration method of how multiple predictions extract the final single prediction, we compare conventional average integration to our EWM method. Performance on the clean condition and the corrupted conditions are labeled as Clean and $mCE$ respectively. Smaller value indicates better

performance.

## 4.2 EVALUATION

From the results from the clean condition of ImageNet, we observed that the loss prediction pipeline has a little and inconsistent impact on the error rate. In most cases, identity transformation is selected by the loss predictor, indicating the data are already in good condition for prediction. For the corrupted data, as the single prediction reduces the $mCE$, cyclic TTA contributes to further improvement. For the target network trained with Augmix, the network has already built strong robustness from the corruption. In this case, both loss prediction pipeline shows minor improvement.

For EWM, while cases of clean data are minorly affected as well, it showed general improvement in the corrupted data. For convention TTA with EWM, as the number of augmentations increases from horizontal flip to 10 crops, the improvement in $mCE$ has increased. This indicates that, as more augmentations are used, more candidates to reflect the uncertainty are available to extract certain and correct answers.

## 4.3 CYCLIC USAGE OF ORACLE-TTA

Kim et al. [Kim et al., 2020] suggested a hypothetically perfect loss predictor named Oracle-TTA to simulate the performance upper bound for the loss prediction pipeline. Oracle-TTA is assumed to be able to accurately predict which transformation is required for the input image to result in the smallest loss value. Hypothetical performance using the Oracle-TTA suggests the potentials in the pipeline. As for comparison, we suggest that the cyclic usage of the Oracle-TTA can further improve the upper bound, for the flexibility in the transformation can provide more optional conditions to the input image for the target network. In appendix D, we compare the upper bound for cyclic TTA to that of our baseline. The performance records show that, with a well trained loss predictor, more rooms for improvements are available as the number of maximum transformation for the cyclic TTA increases.

## 4.4 DISCUSSION

In Appendix B, we illustrate the visual comparison of image conditions between center crop, single iteration, and cyclic iteration methods. We examined that corrupted images can restore some of their features to become closer to their clean condition via multiple iterations of transformations. Additionally, even clean images with ambiguity tend to restore their features, becoming to have similar features to the images of the same class. In our experiment, we have analyzed the results to conclude that the cyclic TTA was more effective on corrupted images with higher corruption

severity and less effective on that of lower corruption severity than our baseline. This is because data with the high severity clearly requires more transformations to restore their features. Moreover, being less well on the lower severity indicates that current cyclic TTA is not well on stopping the iteration at the right time. Without limiting the number of cyclic iteration (the maximum number of iteration), we see that sometimes the image is nearly destroyed, losing much of its features. This indicates that if the loss prediction pipeline is not perfectly well-functional, presence of further unwanted corruption is possible. These indicate that maximum number of iteration parameter should be proportional to the wellness of the loss predictor and additional exit signals should be required to prevent additional unwanted corruptions.

From our experiment, we have observed that well functional loss predictor contributes to even better performance in the cyclic TTA pipeline than in our baseline. On the other hand, the poor performance leads to even more deteriorating result in the cyclic TTA performance, which refers that the accuracy of the loss predictor can lead to more drastic reflection to the performance in the cyclic maneuver. With such observation, it is evident that, the key factor to reach the cyclic Oracle-TTA performance is to train a loss predictor with high accuracy, which involves finding a suitable transformations candidates those are well learnable by the loss predictor and finding a suitable training configuration for the loss predictor.

From the EWM performance difference in the clean and the corrupted condition, we suggest that the measurement of the data uncertainty is more evident in corrupted condition for the given target network. Considering the data uncertainty should be extracted from multiple number of the target network predictions [Malinin et al., 2020], it is possible that the calculated entropy could not have reflected the data uncertainty to an accurate level. From examining the entropy values from "10 crops" case, while the clean data generated relatively uniform entropy values among the augmentations, in the corrupted case, often outlying entropy values was found, which refers to the uncertain augmentations. This indicates that while such data uncertainty reflection could be effective in case of evident distortion in the input image, more precise and accurate measurement of the uncertainty should be required to take the advantage in clean condition of the data.

## 5 CONCLUSION

In this work, we have introduced the cyclic modification of the loss prediction pipeline to implement flexible transformations to the input image and the implementation of EWM for TTA policy. Given that the loss predictor learns the implicit features of the corrupted condition of the image to predict the most suitable transformation, we state that

| Train Time Augmentation | TTA Method | Cost | Average | | EWM (Ours) | |
|---|---|---|---|---|---|---|
| | | | Clean | *mCE* | Clean | *mCE* |
| Standard | Center Crop | 1 | 24.14 | 75.79 | | |
| | Horizontal Flip | 2 | 23.76 | 74.77 | 23.78 | 74.75 |
| | 5 Crops | 5 | 23.57 | 74.37 | 23.47 | 74.22 |
| | 10 Crops | 10 | 23.04 | 73.57 | 23.05 | **73.34** |
| | Single | 1 | 24.15 | 74.14 | | |
| | | 2 | 24.04 | 73.36 | 24.03 | 73.26 |
| | | 3 | 23.84 | 73.23 | 23.85 | 73.08 |
| | Cyclic (Ours) | 1 | 24.15 | **73.69** | | |
| | | 2 | 24.04 | **73.13** | 24.06 | 73.08 |
| | | 3 | 23.81 | **72.74** | 23.81 | 72.68 |
| Augmix | Center Crop | 1 | 22.39 | 65.07 | | |
| | Horizontal Flip | 2 | 22.15 | 64.35 | 22.16 | 64.31 |
| | 5 Crops | 5 | 21.69 | 63.56 | 21.68 | **63.35** |
| | 10 Crops | 10 | 21.56 | 63.05 | 21.49 | **62.76** |
| | Single | 1 | 22.37 | 64.34 | | |
| | | 2 | 22.31 | 63.82 | 22.30 | 63.77 |
| | | 3 | 22.33 | 63.86 | 22.34 | 63.73 |
| | Cyclic (Ours) | 1 | 22.37 | **64.14** | | |
| | | 2 | 22.33 | 63.77 | 22.31 | 63.74 |
| | | 3 | 22.33 | 63.68 | 22.31 | 63.62 |

Table 1: Performance comparison of the previous methods with the proposed method on ImageNet and ImageNet-C. Fourth column indicates averaging for integrating the predictions from multiple augmentations. Fifth column shows the performance with the EWM. Single TTA method refers to the previous method by [Kim et al., 2020]. Cyclic refers to our method. It is bold when either cyclic method or EWM method shows performance improvement of 0.2% or more.

the multiple iterations to find the suitable condition of the corrupted image can be considered as a part of iterative optimization process, and able to restore part of its original quality for network prediction. Our main contribution is to suggest that the cyclic loss prediction pipeline can expand the transformation space of the input image and the upper bound of the loss prediction pipeline via achieving the flexibility of the transformations.

For EWM, we show that direct reflection of data uncertainty could be effective against the corrupted condition of data. As augmentations are given, each of them can contribute with variable weights, for their importance for network prediction are different.

Although we have suggested that such a pipeline holds much potential for performance improvement, there is much gap from the ideal Oracle-TTA performance. Therefore, our future work will be of configuring and training the loss predictor with high performance. As for the transformation candidates, even though we have used a similar set of prede-fined transformations to our baseline, in order to search for a better condition of the input image, it is possible for more transformations with a wider range of magnitude are more suitable. Thus, we plan to experiment with generative models to restore the corrupted condition with respective to the target network. We expect to proceed the transformations without setting the predefined set in future works.

### Acknowledgements

This work was conducted by Center for Applied Research in Artificial Intelligence(CARAI) grant funded by Defense Acquisition Program Administration(DAPA) and Agency for Defense Development(ADD) (UD190031RD).

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
