# OpenReview forum: "Cyclic Test Time Augmentation with Entropy Weight Method"
_auai.org/UAI/2022/Conference — UAI 2022 Poster_

### Official Review · Reviewer_qnwN · 2022-04-09

**Q2(1) Originality/Novelty:** 2
**Q2(2) Significance/Impact:** 2
**Q2(3) Correctness/Technical Quality:** 3
**Q2(6) Clarity Of Writing:** 2
**Q6 Overall Score:** 6
**Q8 Confidence In Your Score:** 4

**Q1 Summary And Contributions:**

The paper proposes cyclical test-time data augmentation (TTA) --- an adaptive method that iteratively selects a sequence of optimal augmentations that will lead to the best loss according to a loss predictor network. The authors also propose another idea ---  Entropy Weight Method (EWM) for weighting the predictions under different augmentations for the case when several alternative augmentations are selected for each image in TTA.

**Q2 Assessment Of The Paper:**

More detailed information regarding each of these aspects is given below:

**Q2(4) Quality Of Experiments (Optional):**

3: Good: The experimental evaluation is adequate, and the results convincingly support the main claims.

**Q2(5) Reproducibility:**

3: Good: Key resources (e.g., proofs, code, data) are available and key details (e.g., proofs, experimental setup) are sufficiently well-described for competent researchers to confidently reproduce the main results.

**Q3 Main Strengths:**

- The proposed idea for cyclical TTA is very logical
- The proposed method is simple, and computationally not very expensive
- Some of the results are good, particularly mCE for a Standard ResNet-50

**Q4 Main Weakness:**

- The writing is often unclear
- The motivation for the EWM is not obvious
- The results on the Augmix-trained network don't look particularly good
- Generally, the empirical evaluation does not convincingly show that the method is useful in practice

**Q5 Detailed Comments To The Authors:**

The paper proposes a very simple extension of [1], where instead of just one augmentation, a sequence of augmentations is picked according to the loss predictor. Several questions:
- How was the loss predictor network trained in the experiments? Is it trained using the corruptions from ImageNet-C or just the original train data?
- How many augmentations do your cyclical policies end up inclluding?
- Can you apply the method of [1] to sequences of augmentations, e.g. consider all the possible augmentation sequences of length 5? Would that be feasible computationally? I imagine, it should work at least as well as the proposed method.

The propose EWM weighting method is not motivated very well in my opinion. In particular, regular averaging of the softmax class probabilities is already taking uncertainty into account. Indeed, imagine  for one of the augmentations the network is 99% confident in class 0, and for the second augmentation it's 50% confident in class 1. Then, the prediction will be class 0. So why do you think we need an extra confidence-based weight for the different augmentations?

The empirical results look good on the standard network, where the cyclical TTA with both averaging types significantly outperforms the standard fixed augmentation baselines. However, on the Augmix-trained network the results are not as good. In particular, 5 crops and 10 crops fixed augmentation methods perform better than the proposed adaptive cyclical TTA. Consequently, the empirical results are mixed.
- Why do you think the performance on Augmix-trained network is not good?
- Are there other problems that could be interesting to consider in order to show that the proposed method works well? Perhaps, you could consider CIFAR datasets, e.g. CIFAR-100 with a WideResnet and ResNext, following the setting considered in [1]?

Finally, the writing is not good, often confusing. The structure of the paper is reasonable, but the phrasing of the sentences is often hard to parse. Just as an example:
> In other words, the loss predictor tells of what transformation would result in the best outcome for the target network, in scores of minimum the better.
I don't get what it means, and a lot of sentences in the paper are like this. I ask the authors to carefully go through the paper and try to make sure that all the sentences are clearly phrased.

**References**

**[1]** *Learning Loss for Test-Time Augmentation*
Ildoo Kim, Younghoon Kim, Sungwoong Kim

**Q7 Justification For Your Score:**

The paper proposes a very simple and logical idea, which I believe could be highly practical. However, the empirical evaluation is not entirely convincing, as the method works well for one model, and not as well for the other.

**Q9 Complying With Reviewing Instructions:**

1: Yes.

---

### Official Review · Reviewer_c8J3 · 2022-04-12

**Q2(1) Originality/Novelty:** 3
**Q2(2) Significance/Impact:** 2
**Q2(3) Correctness/Technical Quality:** 2
**Q2(6) Clarity Of Writing:** 3
**Q6 Overall Score:** 5
**Q8 Confidence In Your Score:** 3

**Q1 Summary And Contributions:**

This paper investigates a type of data argumentation method, called test time augmentation. By introducing cyclic optimization process and entropy weight method, their proposed EWM can improve the instance-level TTA with a more flexible magnitude, and achieve better ability. Some experiments are conducted to verify the advantage of their approach.

**Q2 Assessment Of The Paper:**

More detailed information regarding each of these aspects is given below:

**Q2(4) Quality Of Experiments (Optional):**

2: Fair: The experimental evaluation is weak: important baselines are missing, or the results do not adequately support the main claims.

**Q2(5) Reproducibility:**

3: Good: Key resources (e.g., proofs, code, data) are available and key details (e.g., proofs, experimental setup) are sufficiently well-described for competent researchers to confidently reproduce the main results.

**Q3 Main Strengths:**

+ Sound technical contribution. They assemble powerful techniques to improve the ability of  test time augmentation. The introduce of each   technique is reasonable and useful.
+ Good presentation. It is good to follow.

**Q4 Main Weakness:**

- It is somehow trivial to achieve better performance because some advanced techniques are beneficial to proposed approach.
- Ablation study is needed to verify which of cyclic learning or entropy weight method is more significant.
- More experiments on other datasets are also needed.

**Q5 Detailed Comments To The Authors:**

The interaction of cyclic learning or entropy weight method should be discussed.
And more experiments are needed to verify proposed method, including more datasets, more ablation, more baselines.

**Q7 Justification For Your Score:**

The techniques are sound, but experiments are inadequate.

**Q9 Complying With Reviewing Instructions:**

1: Yes.

---

### Official Review · Reviewer_Rh4m · 2022-04-27

**Q2(1) Originality/Novelty:** 3
**Q2(2) Significance/Impact:** 2
**Q2(3) Correctness/Technical Quality:** 3
**Q2(6) Clarity Of Writing:** 3
**Q6 Overall Score:** 5
**Q8 Confidence In Your Score:** 3

**Q1 Summary And Contributions:**

The paper studies Test Time Augmentation for improving test time performance. In particular, it proposes to use cyclic search for suitable transformations as well as an entropy weight method. The experimental result shows that it improves standard image classification as well as robustness to image corruptions.

**Q2 Assessment Of The Paper:**

More detailed information regarding each of these aspects is given below:

**Q2(4) Quality Of Experiments (Optional):**

2: Fair: The experimental evaluation is weak: important baselines are missing, or the results do not adequately support the main claims.

**Q2(5) Reproducibility:**

3: Good: Key resources (e.g., proofs, code, data) are available and key details (e.g., proofs, experimental setup) are sufficiently well-described for competent researchers to confidently reproduce the main results.

**Q3 Main Strengths:**

+ The usage of cyclic search for suitable transformations seems novel and interesting.

+ The experiment results are good compared to existing TTA methods

+ The overall writing is clear and easy to follow

**Q4 Main Weakness:**

- There's a missing line of related research topic, which is test-time training. How does the proposed method (TTA) perform compared to TTT-based methods?

- The evaluation is not that thorough. Though both standard image classification as well as robustness to image corruptions are tested, only one dataset is used, which might cause the problem of whether it generalizes to other datasets.

- What is the inference computation cost of Cyclic method? How does it compare to standard inference, as well as other TTA methods?

**Q5 Detailed Comments To The Authors:**

Please refer to the previous sections.

**Q7 Justification For Your Score:**

I think in general the paper has merits. I would like to see other reviews and the author response on my questions.

**Q9 Complying With Reviewing Instructions:**

1: Yes.

---

### Decision · Program_Chairs · 2022-05-15

**Decision:**

Accept (Poster)

**Comment:**

Meta Review: All reviewers had a (mildly) positive view about the paper, so acceptance is merited.